# Nanotechnology-Based siRNA Delivery Systems to Overcome Tumor Immune Evasion in Cancer Immunotherapy

**DOI:** 10.3390/pharmaceutics14071344

**Published:** 2022-06-25

**Authors:** Kaili Deng, Dongxue Yang, Yuping Zhou

**Affiliations:** 1Department of Gastroenterology and Hepatology, The Affiliated Hospital of Medical School, Ningbo University, Ningbo 315020, China; dkl1640@126.com (K.D.); 17775498861@163.com (D.Y.); 2School of Medicine, Ningbo University, Ningbo 315021, China; 3Institute of Digestive Disease of Ningbo University, Ningbo 315020, China

**Keywords:** small interfering RNA, immune evasion, nanoparticles, tumor microenvironment, immunotherapy

## Abstract

Immune evasion is a common reason causing the failure of anticancer immune therapy. Small interfering RNA (siRNA), which can activate the innate and adaptive immune system responses by silencing immune-relevant genes, have been demonstrated to be a powerful tool for preventing or reversing immune evasion. However, siRNAs show poor stability in biological fluids and cannot efficiently cross cell membranes. Nanotechnology has shown great potential for intracellular siRNA delivery in recent years. Nano-immunotherapy can efficiently penetrate the tumor microenvironment (TME) and deliver multiple immunomodulatory agents simultaneously, which appears to be a promising method for combination therapy. Therefore, it provides a new perspective for siRNA delivery in immunomodulation and cancer immunotherapy. The current advances and challenges in nanotechnology-based siRNA delivery strategies for overcoming immune evasion will be discussed in this review. In addition, we also offer insights into therapeutic options, which may expand its applications in clinical cancer treatment.

## 1. Introduction

Cancer ranks as a one of the leading causes of death, with a substantial barrier to extend life expectancy worldwide [1]. In the past 30 years, despite the significant progress has been made for the mainstream cancer treatment methods including surgery, chemotherapy, radiotherapy, or their combinations, increasingly empirical consensus has shown that the failure of these standard treatments is largely attributable to inborn or acquired therapeutic resistance [2]. Targeted therapy, gene therapy, and immunotherapy are newly established treatment techniques that have expanded cancer treatment options and enabled individualized medicine [3,4].

Unlike the targeted therapy or gene therapy, the effect of immunotherapy is achieved by strengthening an anticancer immune response [5]. Although cancer immunotherapies have demonstrated promise in several preclinical models, a limited proportion of patients react to therapy [6]. The growth and progression of cancer are governed by immune cells [7]. However, connections between immune system components and tumor cells in TME frequently result in immune evasion, which enables cancer cells to evade killing by immune cells [8]. Selecting tumor variants resistant to immune effectors and the continual formation of an immunologically suppressive milieu within the tumor contribute to immune evasion [9]. Indeed, it has been demonstrated that the recruitment and expansion of immunosuppressive cells, such as regulatory T cells (Tregs), tumor-associated macrophages (TAMs), and myeloid-derived suppressor cells (MDSCs), are related to tumor aggressiveness and poor prognosis [10]. For effective tumor control, therapeutic treatments focus on modifying the TME and/or reducing immunosuppression, as well as counteracting insufficient tumor immunogenicity and antigenicity [11]. Immunomodulatory substances such as cytokines, adjuvants, and monoclonal antibodies are utilized to modify the TME and enhance antitumor immunity [12]. Despite the preliminary success of immunotherapeutic agents, some patients do not benefit from them. In addition, a substantial number of people have been reported to suffer serious side effects, such as gastrointestinal, hematologic, and endocrine illnesses, arthritis, dermatitis, neuropathy, and acute renal injury [13,14].

SiRNA is essential for efficiently suppressing multi-drug-resistance-related protein, inhibiting tumorigenesis protein/suppressor mutations, and activating tumor-associated immune cells [15,16]. Applying siRNA methodology in cancer treatment, particularly beyond liver targets, has been intensively researched. However, as a significant drawback, unmodified siRNA is unstable in the bloodstream and is prone to immunogenicity. Therefore, it cannot readily enter cells for crossing membranes [17]. The lack of unharmful and convenient transport systems is a fundamental obstacle to realize the vast potential of siRNA-based therapies [18]. As shown in Figure 1, chemical modifications and delivery systems have been established to safely transport siRNA to its location of action [19]. At present, the nanoparticles that are used as loading systems for antitumor drugs usually have a size of 1–100 nm, including nano-liposomes, nano-polymers, nano-gene carriers, nano-inorganic materials, and other drug carriers [20,21]. The active targeting can be achieved by the targeted transportation modes of antitumor drug nanocarriers, including passive transportation, active transportation, and physical and chemical transportation [22,23]. The tumor microenvironment (TME) comprises a range of stimuli, such as enzymes, mildly acidic pH, hypoxia, and GSH, which can help with the development of stimuli-responsive nanoscale drug delivery systems (SRNDs). Because these SRNDs preserve their stealth characteristics in the normal physiological environment, they are sensitive and release their contents when homing in on particular lesions [24,25]. Therefore, in comparison to conventional medications, nanotechnology-based siRNA delivery systems can alter the immunosuppressive environment by targeting primary components in the TME, hence enhancing the efficiency of cancer immunotherapy [15]. Furthermore, nano-siRNA may increase retention time and enable targeted delivery, thus minimizing toxicity. The application of nano-siRNA as a drug delivery strategy has been extensively explored [18]. In general, nanotechnology-based siRNA delivery systems have predominantly evolved as novel techniques for modifying the immune system in two ways [26]: initially, targeting and/or removing immunosuppressive cells or pathways that implicated in tumor immune evasion; secondly, activating cytotoxic T lymphocytes (CLTs) to generate immunogenic cell death (ICD).

In this review, we summarize the immune evasion mechanisms that lead to the failure of cancer treatment modalities. In addition, we also highlight the current advances and challenges in nanotechnology-based siRNA delivery strategies for overcoming immune evasion.

## 2. Immune Evasion Mechanisms in Tumor

Cancer immune surveillance is a crucial process that enables the immune system to monitor, identify, and eliminate tumor cells during the early stages of carcinogenesis [27]. This process has three fundamental steps called elimination, balance, and escape [28]. Several investigations have revealed that the immune system has the potential to identify and eliminate tumor cells via the recruitment of innate and adaptive effectors [29]. In the context of antitumor responses, innate immunity promotes fast and non-specific responses, while adaptive immunity is more specific [30]. The first line of protection is the innate immune system, which detects microbes and endogenous danger signals by recognizing damage-associated molecular patterns (DAMPs) or pathogen-associated molecular patterns (PAMPs) through host pattern recognition receptors (PRRs), such as Toll-like receptors (TLRs) and nuclear receptors [31]. It has been reported that immune cells including NK cells, monocytes/macrophages, and neutrophils can contribute to the elimination of tumor cells through indirect or direct mechanisms, such as the generation of antitumor chemicals or antibody-dependent cellular cytotoxicity (ADCC) [32]. Professional antigen-presenting cells (APCs), such as dendritic cells (DCs), act as bridges between innate and adaptive immunity, which connect activating T cells (CTL or helper T cells) and B cells [33]. DCs move to neighboring lymph nodes (LN) during maturation, where they present tumor antigens and activate tumor-specific CD4+ and CD8+ T cells [34]. Then, these tumor-specific T lymphocytes will migrate to the tumor location and aid in its elimination.

Growing tumors can defy immune responses by excluding or hiding from immune cells through intrinsic pathways [35] and extrinsic pathways [36], resulting in the formation of a favorable environment for tumor development.

### 2.1. Intrinsic Mechanisms Mediating Immune Evasion

Tumor cells have evolved a variety of mechanisms to evade immune responses. Indeed, these escape mechanisms are selected by the cancer cells after a period of engagement with the immune system [37]. Multiple tumor cells exhibit reduced levels of MHC class I, downmodulation of tumor antigens, and weak immunogenicity, which are highly associated with tumor immune evasion (Figure 2).

To elicit an efficient antitumor response, antigen presentation requires two independent steps. First, cancer antigens must be picked up by DCs and cross-presented to CD8^+^ T cells for priming. Second, the tumor needs to directly deliver the antigens for detection and killing by primed CD8^+^ T cells. Tumors employ a variety of mechanisms to avoid detection by the immune system during both stages [38]. For example, tumor-specific CD8^+^ T cells recognize and activate tumor antigens, and the particular killing of tumor cells is dependent on the T cell receptors (TCR) specifically recognizing and attaching to MHC-I-peptide complexes [39]. To successfully evade immune identification, tumor cells can change in the interaction between MHC molecules and antigenic peptides, which have an effect on the TCR recognition of MHC–antigenic peptide complexes [40,41]. Additionally, certain tumors lack pre-existing tumor T cell infiltration, allowing them to elude immunosurveillance as a result of low tumor antigen expression levels [41,42] The low expression levels of tumor antigen result in poor APC recruitment and activation, hence impeding the beginning of an efficient immunological response [43].

Immune surveillance can detect and kill tumor cells at the early stages of cellular transformation [44]. However, since tumor cells vary in their immunogenicity, tumor cells with strong immunogenicity can induce an effective antitumor immune response and are easily eliminated, whereas tumor cells with weak immunogenicity can corrupt the host’s antitumor immune response and arise, survive, and grow [45]. Immune selection enables tumors cells with low immunogenicity to evade immune monitoring and proliferate preferentially [46]. Additionally, studies indicate that the lack of immunogenicity may be attributable to the improper expression or absence of costimulatory molecules on tumor cells [47].

Downregulation of MHC-I on tumor cells, a widely used tactic by tumor cells to evade specific immune responses, may be linked to coordinated silencing of antigen-presenting machinery genes [48,49,50]. Direct or indirect mutations in genes encoding antigen processing or presentation components, such as proteasome subunits and endoplasmic reticulum peptide transporters, can inhibit the expression of MHC-I in malignant cells [50,51]. Evading immune recognition by tumor cells through downregulating the expression of MHC-I compromises the ability of tumor associated antigen (TAA)-specific CTLs to kill cancer cells but boosts recognition and killing by NK cells when total MHC I levels fall below a threshold [52].

### 2.2. Extrinsic Mechanisms Mediating Immune Evasion

Immune cells, fibroblasts, endothelial cells, inflammatory cells, and lymphocytes make up the TME, along with extracellular matrix (ECM), vasculature, and chemokines [53,54]. Immune cells are involved in innate and adaptive immune responses. The adaptive immune system is capable of destroying tumor cells precisely and is perceived as the most efficient means of removing malignancies [55]. The TME is distinguished by an atypical tumor vasculature that appears and functions abnormally, resulting in a hypoxic condition [56]. In addition, hypoxia in the TME can alter abilities of the normal microenvironment, increase the growth of the tumor, and limit the therapeutic effects [57]. Thus, connections between tumor cells and components of the TME influence tumor growth, metastasis, and clinical outcome. TME has shown a significant effect on medication penetration and function, as well as being connected with drug resistance and low response rates [58]. Effective cancer remission is mediated by immunotherapy that induces sufficient immune responses. Unfortunately, the interaction between the immune system and tumor cells leads to increased immunosuppression in the TME and reduced immunogenicity of tumor cells, allowing tumor cells to resist immune elimination [59,60]. Malignant cells secrete a variety of chemokines and cytokines that encourage immune cells such as Tregs, TAMs, and MDSCs to infiltrate into the tumor [61]. Due to the regional production of immunosuppressive cytokines, chemokines, and growth factors, as well as their interaction with TME components, these immune cells are recognized as cancer immune suppressors (Figure 3) [62]. To this end, researchers interested in cancer immunotherapy have placed a premium on TME modulation.

Tregs are a T cell subset with regulatory immunological capabilities that act as immunosuppressive factors in the TME and encourage tumor progression [63]. Chemokines in the TME have been demonstrated to attract naturally existing Tregs from the thymus, bone marrow, lymph nodes, and periphery via the Treg receptor CCR4, hence weakening immunity [64].

Tregs largely contribute to the immune evasion of tumor cells by inhibiting effector T cells and DCs via many mechanisms [65]. First, Tregs can secrete perforin (PFN) and granzyme B (GzmB), which directly act on effector cells to promote apoptosis [66]. In addition, Tregs could secrete immunosuppressive cytokines such as TGF-β, IL-10, and IL-35, which attach to immunological cells and inhibit the immune system [67,68]. Furthermore, numerous surface receptors, incorporating cytotoxic T lymphocyte-associated antigen-4 (CTLA-4) [68], programmed cell death protein 1 (PD-1), lymphocyte-activation gene 3 (LAG-3), T cell immunoglobulin, and mucin-containing molecule 3 (TIM-3) [69,70], as well as upregulation of indoleamine 2,3-dioxygenase 1 (IDO) and activation of CD28 family-induced costimulatory molecules (ICOS), contribute to the immunosuppressive function of Tregs which promotes the secretion of inhibitory cytokines [71]. As a corollary, it is envisaged that eliminating Tregs or inhibiting their immunosuppressive actions will restore the antitumor effects of immunotherapies [72].

TAMs, which are myeloid-derived and tissue-resident macrophages prominent in the microenvironment of solid tumors, become immunosuppressive when they interact with tumor cells [73]. They have been shown to be critical ingredients of the TME and to encourage tumor growth. TAMs seem to be synonymous with M2 macrophages due to that they share M2 phenotypic traits, including boosting angiogenesis, reducing inflammation, and matrix remolding [74].

TAMs are classified into two subtypes based on their cellular context: tumor-suppressive M1 macrophages and tumor-promoting M2 macrophages [75]. Preclinical and clinical research has revealed that TAMs are largely M2 phenotype, which participate in the tumorigenesis and cancer progression [76]. M1 macrophages have microbicidal and anticancer capabilities [77]. In comparison, anti-inflammatory cytokines IL-4, IL-10, and IL-13 are secreted by M2 macrophages, as well as a reduction in reactive oxygen species (ROS), NO, and TNF production, which are responsible for the anti-inflammatory effect [74,78].

The predominance of M2-like TAMs in the TME contributes to tumor immune evasion and chemoresistance. Through the secretion of soluble substances such as IL6 and PGE2, tumor cells directly stimulate the M2 polarization of macrophages [78,79]. M2-macrophages induce a Th2-type response, which inhibits proinflammatory stimuli. TAMs exert their immunosuppressive activity in a variety of ways, including the release of IL-10 and TGF-β, activation of IDO, generation of prostaglandins, and upregulation of the programmed death-ligand 1(PD-L1), programmed death-ligand 2 (PD-L2), and VISTA checkpoints [74].

MDSCs, as a heterogeneous population of immature myeloid cells, have a high level of anti-T cell activity [80]. MDSCs accumulate in different kinds of tumors tissue, where they enhance tumor invasion, angiogenesis, and metastasis while inhibiting antitumor immunity [81]. Thus, MDSCs provide a vital function in helping tumor cells to evade immune monitoring, thereby promoting tumor proliferation and progression [81]. The accumulation of MDSCs in TME is primarily determined by two modulation schemes. To begin, tumor cells induce MDSCs generation and recruitment via the secretion of stem cell factor (SCF), granulocyte macrophage colony-stimulating factor (GM-CSF), granulocyte colony-stimulating factor (G-CSF), vascular endothelial growth factor (VEGF), and macrophage colony-stimulating factor (MCSF) [82,83,84]. The second class of signals consists of cytokines and chemokines such as IL-4, IL-6, IL-1, and CXCL1, that are mostly produced by the tumor stroma, which induce the suppressive activity of MDSCs via NF-κB, STAT1, and STAT6 [83,84,85]. Intra-tumoral MDSCs can impair antitumor responses by upregulating pathways involved in the production of arginine, ROS, and nitric oxide (NO), as well as by secreting TGF-β, which can dampen effector T cells directly or indirectly to promote immune evasion [86].

## 3. Targeting Immunosuppressive Cells in the TME with Nanotechnology-Based siRNA Delivery Systems

Strategies for TME regulation have received considerable attention in cancer immunotherapy. Unlike conventional treatment, nanocarriers with distinct physical properties and complex structures can successfully penetrate TME and distribute to specific components [87]. At the same time, nanoparticles can increase the retention duration and distribute drugs more precisely, thereby lowering toxicity [88]. Additionally, nanoparticles have the potential to alter the immunosuppressive milieu within TME by targeting the key components [89]. The deformed blood vessels and rapid development of tumor cause hypoxia, which result in immunosuppression in TME through the accumulating immunosuppressive cells such as Tregs and MDSCs and secreting immunosuppressive factors such as VEGF and TGF-β. This substitute impairs the functions of DCs, converting macrophages to the tumor-promoting M2 phenotype, and resulting in immune evasion [90]. Nanoparticles can target certain components in the TME and convert them to an immune supporting state, hence increasing the efficacy of cancer immunotherapy [91]. Nanotechnology-based siRNA delivery systems have been extensively investigated as potential novel therapy modalities for a variety of cancers over the last few decades [92]. When modified with various ligands, nanoparticles function as efficient drug delivery systems that can selectively target TME components such as Tregs, macrophages, MDSCs, fibroblasts, tumor vasculature, tdLNs, and the hypoxic state. By encapsulating siRNA in nanoparticles, we can preserve siRNA from degradation and efficiently distribute it to tumor cells [93]. Nanotechnology-based siRNA delivery systems have been utilized extensively for in vivo gene silencing and tumor therapy [94]. Here, we will discuss the specific siRNA delivery systems used to target important immunosuppressive cell populations or pathways.

### 3.1. Targeted Delivery of siRNA to Tregs

Depletion of Tregs or inhibiting their immunosuppressive actions could restore the antitumor activity of CTLs, hence preventing the development of tumors (Figure 4) [95]. Unfortunately, clinical implementation of Treg-depleting methods remains unsatisfied due to the paucity of Treg cell-specific markers and small molecules that specifically target Treg functions [96]. It has been reported that covalently linking siRNA to an aptamer (apt) that specifically binds cytotoxic T lymphocyte-associated antigen 4 CTLA4 (apt) enables gene silencing, as well as CTLA4-expressing malignant T cells. CTLA4 (apt) link to a STAT3-targeting siRNA (CTLA4 (apt)-STAT3 siRNA), resulting in STAT3 suppression and the activation of tumor antigen-specific T cells. Additionally, CTLA4(apt)-STAT3 siRNA demonstrated a substantial inhibiting impact on tumor development and metastasis in a variety of mouse tumor models [97]. Recent improvements in nanoparticles have facilitated the creation of novel Treg-depleting strategies. For example, Wang et al. constructed nanoparticles to deliver CTLA-4-siRNA (NPsiCTLA-4), which are used to transfer specific siRNAs to T cells, therefore promoting the activation and proliferation of T cells. They created siCTLA-4-encapsulated nanoparticles using a biocompatible and biodegradable poly (ethylene glycol)-block-poly (D, L-lactide) (PEG5 k-PLA11 k) copolymer and N-bis(2-hydroxyethyl)-N-methyl-N-(2-cholesteryloxycarbonyl aminoethyl) ammonium bromide (BHEM-Chol). The findings confirmed that the nanocarrier delivery system could deliver CTLA-4-siRNA to CD4+ and CD8+ T cell subsets at tumor sites, while decreasing the proportion of suppressor Tregs among tumor-infiltrating lymphocytes (TILs), thereby increasing the antitumor immune response of tumor-infiltrating T cells [98]. To investigate the antitumor effects of various combined strategies, Zhang et al. created several spherical nucleotide nanoparticles (SNPs) that were loaded with CTLA-4-siRNA aptamer (cSNPs), PD-1 siRNA (pSNPs) or both (hybrid SNPs, or hSNPs). The findings showed that hSNPs could promote considerably higher antitumor immune responses than a combination of pSNPs and cSNPs (pSNPs and cSNPs),through regulating the immune suppressive function of both Tregs and TIM3+ exhausted-like CD8 T cells [99]. Overall, utilizing nano-siRNA delivery systems to achieve Tregs depletion may be a realistic strategy.

### 3.2. Targeted Delivery of siRNA to TAMs

Repolarizing or eradicating regulatory TAMs is a strategy that may be plausible for TAMs-targeted cancer immunotherapy (Figure 5) [100]. TAMs are an attractive population cell for such endeavors due to their plasticity, which allows them to transition into M1-like phenotype from M2-like phenotype. Additionally, macrophages readily internalize particles due to their natural phagocyte status, significantly increasing uptake efficiency [73]. Therefore, modifying the polarization of TAMs may impact their function, which has garnered much attention [101]. Nanoparticles are capable of delivering medications specifically to TAMs and modulating their polarized states, providing a promising strategy for cancer immunotherapy [102]. Thus, inhibiting macrophage infiltration, preventing the formation of M2-like TAMs, repolarizing majority M2-like TAMs to M1-like TAMs, or epigenetically suppressing the release of M2-like TAM-induced factors in the TME may all be plausible for TAMs-targeted cancer immunotherapy [103]. Toll-like receptor (TLR) agonists, such as CpG oligodeoxynucleotides (CpG ODNs), can activate antitumor macrophages, but their in vivo efficacy is limited due to the lack of effective delivery methods [104]. Naked CpG ODNs cannot permeate cell membranes and are rapidly removed by nucleases, posing a risk of an inflammatory reaction in the serum when given systemically [105,106]. It has been demonstrated that nanoparticles can deliver TLR agonists to TAMs, selectively concentrating in tumors and macrophages, eventually triggering TLR signaling and M1 polarization [107]. Therefore, a self-assembled nucleic acid system was used to construct an efficient siRNA and CpG ODNs delivery system (CpG-siRNA-tFNA). The combination of CpG-siRNA-tFNA efficiently polarized TAMs toward the M1 phenotype, which resulted in an enhanced production of proinflammatory cytokine and activation of the NF-κB signaling pathway, therefore eliciting robust anticancer immune responses. Furthermore, the combination of CpG-siRNA-tFNA increased antitumor activity without causing systemic toxicity in a mouse model of breast cancer xenograft [108].

Apart from altering the polarization of TAMs, another tactic is to impair their survival and function [109]. A recent study [110] used dual-targeting nanoparticles to deliver siRNA to M2-like TAMs to develop a molecular-targeted cancer immunotherapy strategy. A biocompatible fusion peptide-functionalized lipid nanoparticle with a dual-targeting entity for specific M2-like TAM binding, a sub-30 nm size for efficient solid tumor penetration [111], and stable loaded siRNA for systemic transport is the main component of this method [112] Qian et al. developed a new M2-like TAM-targeting core–shell fluorescent lipid nanoparticle (M2 NP), whose structure and function were controlled by α-peptide (a scavenger receptor B type 1 (SR-B1) targeting peptide) linked with M2 pep (an M2 macrophage binding peptide) [110]. This nanocomplex, which is encoded with anti-colony-stimulating factor-1 receptor siRNA on the M2 NP, is more selective and has a higher affinity for TAMs than other macrophages. Compared to control groups, M2 NP-based siRNA administration significantly reduced M2-like TAMs (52%), limited tumor size, and prolonged survival. Additionally, this molecularly targeted method decreased the production of immunosuppressive IL-10 and TGF-β, and enhanced the expression immunostimulatory cytokines and infiltration of CD8^+^ T cells in the TEM. Furthermore, the siRNA-carrying M2 NPs decreased the expression of PD-1 and Tim-3 on invading CD8+ T cells and increased the production of IFN-γ, showing that T cell immunological activity has been restored [110]. Thus, impairing the survival of TAMs with delivery of siRNA provides a potential strategy for clinical application of molecular-targeted cancer immunotherapy.

### 3.3. Targeted Delivery of siRNA to MDSCs

MDSCs are the primary facilitators of tumor survival and antitumor immune suppression, as evidenced by the secretion of Th2-type cytokines that lead to the development of an immunosuppressive TME [113]. MDSCs are involved in tumor immune evasion [114]. Hence, the concept of reprogramming or reducing MDSCs as a therapeutic strategy is evolving as a unique method for the continual improvement of existing cancer immunotherapies (Figure 5) [115]. Granulocytic MDSCs develop in patients with prostate cancer as the disease progresses. Hossain et al. [116] established the feasibility of targeting MDSCs using STAT3 siRNA to ameliorate arginase-dependent inhibition of T cells. The team developed an innovative technique for precisely silencing genes in Toll-like Receptor-9 (TLR9) positive myeloid cells utilizing CpG-siRNA conjugates. Human granulocytic MDSCs express TLR9 and rapidly internalize naked CpG-STAT3 siRNA, effectively suppressing the expression of STAT3. The inhibition of STAT3 abolishes the immunosuppressive effects of MDSCs generated from patients’ effector CD8^+^ T cells. These results demonstrate the viability of employing TLR9-targeted STAT3 siRNA delivery to reduce MDSC-mediated immunosuppression [116].

With the goal of achieving potent antitumor immunity, an immunochemotherapy regimen based on a redox-responsive nano-assembly (R-mPDV/PDV/DOX/siL) was designed. R-mPDV/PDV/DOX/siL was self-assembled by three synthesized amphiphilic polymers, including mPEG-DA-PVL (mPDV), RGD-mal-PEG-DA-PVL (R-mPDV), and PAMAM-DA-PVL (PDV), along with DOX encapsulated in core and LDHA siRNA (siL) compressed by PAMAM. These redox-responsive carrier materials were synthesized by conjugating hydrophobic polyester material PVL with hydrophilic mPEG or G2 PAMAM using 3, 3′-dithiodipropionic acid (DA) as the GSH-responsive linkage. This redox-responsive nano-assembly combined the strategies of inhibiting cytokine-mediated MDSC recruitment via LDHA silencing and enhancing tumor immunogenicity via anthracycline (DOX)-induced ICD effects. After egressing from endosomes/lysosomes, R-mPDV/PDV/DOX/siL is disintegrated by GSH-induced DA cleavage, enabling rapid drug release and very effective LDHA silencing. Reduced LDHA expression inhibits the production of G-CSF and GM-CSF cytokines, leading to inhibited MDSC recruitment and strengthened antitumor immunity [117].

Overall, these investigations suggest that targeted MDSCs reduction with siRNA delivery systems may be used as monotherapies or in conjunction with T cell-boosting drugs to enhance therapeutic efficacy.

## 4. Targeted Delivery of siRNA to Checkpoint Inhibitors

Immune checkpoint blockade therapies have been developed as effective treatment methods for a range of tumor types in recent years [118]. Checkpoint inhibitors inhibit the expression of specialized proteins on the surfaces of tumor cells and immune cells that are involved in immune evasion via the activation of certain signaling pathways [119]. PD-L1 is a type of cell membrane protein that is associated with tumor growth. By inhibiting the ligand–receptor interaction of PD-1/PD-L1, it is possible to prevent the activation of the PD-1/PD-L1 signaling pathway, reversing T cell fatigue and preventing immunological responses in the tumor microenvironment (Figure 6) [120]. Therefore, monoclonal antibodies against PD-1/PD-L1 have been the subject of numerous preclinical and clinical studies [121,122].

Since overexpression of PD-L1 on the surface of tumor cells contributes to immune evasion, limiting PD-L1 expression has been proven to be an efficient method for facilitating immune system activation and inhibiting tumor development. Immune checkpoint inhibition has been accomplished in this manner by preloading nanocarriers with siRNA directed against specific checkpoint inhibitory pathways. Li et al. proposed using hybrid nanoparticles composed of polyethylene glycol-polylactic acid (PEG-PLA) and the cationic lipid BHEM-Chol to deliver anti-CTLA-4 siRNA in a recent study [98]. Anti-CTLA-4 siRNA-loaded nanoparticles were efficiently internalized by TILs (4–6%) after systemic distribution, resulting in tumor growth reduction and longer survival in B16 F10 melanoma-bearing mice. Teo and colleagues constructed folic acid (FA)-functionalized polyethyleneimine (PEI)-based nanoparticles to facilitate CTL-mediated killing of cancer epithelial cells (SKOV-3) through the release of siRNA that targets the PD1/PD-L1 pathway [123]. These findings reveal that siRNA-targeted silencing of PD-L1 can sensitize cancer cells to T cell-mediated death. Likewise, Roeven et al. used a lipoplex vector made up of SAINT-RED:DOPE liposomes precomplexed with anti-PD-L1 siRNA to achieve efficient and long-term PD-L1 silencing without compromising DC maturation or viability [124]. Together with other previously reported strategies for checkpoint inhibition, the PD-L1-siRNA loaded lipoplexes represent an innovative approach for improving the efficacy of antigen-specific immunotherapy, such as vaccines and ICD-inducing therapies, without risking treatment safety and tolerability. A recent work referred to as NPs@apt developed a tailored siRNA delivery system for transfecting PD-L1 siRNA into A549 cells to inhibit tumor immune evasion. The result revealed that PD-L1 siRNA was administered precisely to A549 cells, resulting in PD-L1 gene silencing, T cell activation, and reduction of tumor cell proliferation [125]. Additionally, this work developed an innovative method for targeted siRNA transfection to enhance antitumor immunity. These studies have shown that encapsulating siRNA in a nanoparticle can improve its stability and targetability, potentially addressing the issue of undesirable toxic effects associated with checkpoint blockade therapy.

## 5. Targeting Tumor Cells with Nano-siRNA: ICD-Inducing Strategies

Recent research suggests that traditional cancer therapies can be curative not only through directly killing malignant cells, but also through the induction of innate and adaptive antitumor responses, leading to the establishment of a new class of antitumor treatments known as ICD-inducing strategies [126]. ICD-inducing therapies are characterized by their capacity to enhance APC activation, priming of tumor-specific CD8+ T cells, and recruiting immune cells via the production of tumor antigens PAMPs and/or DAMPs, including calreticulin (CRT), heat shock proteins (HSPs) 70 and 90, and high-mobility group box 1 (HMGB1) (Figure 7) [127,128]. ICD has recently received considerable interest since it can be triggered by a variety of stimuli and anticancer treatment techniques, such as chemotherapy, radiotherapy, UVC irradiation, oncolytic viruses, and photodynamic therapy (PDT) [127,129,130,131]. ICD might differ in terms of DAMP profile in response to diverse stimulus and has also been associated with various cell death modalities, including apoptosis, necroptosis [132,133], and ferroptosis [134].

Using siRNA-based nanocarriers to activate ICD is appearing as an attractive fresh approach for producing a strong antigen-specific immune response, with potential to improve the efficiency and reliability of traditional ICD treatments, such as chemotherapies and PDT [135]. Several studies have demonstrated that the immunogenicity of these ICD-inducing monotherapies can be further enhanced by combining ICD-inducing monotherapies with other TME-targeting immunotherapies, [136]. For instance, an exosome-based dual-delivery biosystem has been demonstrated to enhance pancreatic ductal adenocarcinoma (PDAC) immunotherapy [136]. The delivery system is composed of exosomes derived from bone marrow mesenchymal stem cells, electroporation-loaded galectin-9 siRNA, and OXA prodrug used as an ICD inducer. Bone marrow mesenchymal stem cells (BM-MSCs) exosomes dramatically enhance tumor targeting efficacy, resulting in increased drug accumulation at the tumor location. The combination therapy (iEXO-OXA) induces antitumor immunity via tumor-suppressive macrophage polarization, recruitment of cytotoxic T cells, and downregulation of Tregs, and achieves considerable therapeutic efficacy in PDAC treatment [137].

PDT is a new non-invasive light-triggered therapeutic approach that has been clinically authorized and utilized to treat a range of malignancies [138]. PDT has the ability to increase the immunogenicity of tumors by stimulating the production of tumor antigens and triggering the release of a variety of DAMPs [139]. Nevertheless, due to tumor hypoxia and immune evasion, PDT is inefficient, thus reducing their photosensitizing efficiency and, consequently, the therapeutic effect [140]. Nanoparticles have successfully circumvented these constraints by delivering PDT in combination with immunotherapy [141]. Indocyanine green (ICG), which has notable near-infrared (NIR) optical properties within the optimal biological window for biomedical applications, has been extensively studied for NIR-fluorescence-guided imaging [142], as well as its great potential in PDT and photothermal therapy (PTT) due to deep permeation into tissues [143]. MnO_2_- and CaCO_3_-based nanomaterials have received a lot of attention in recent years for their ability to function as carriers for targeted medication delivery to regulate the TME inside solid tumors [144,145]. Herein, a nanoplatform of Mn@CaCO_3_/ICG@siRNA was designed and fabricated. The walnut-shaped MnO_2_ nanoparticles were generated by reducing potassium permanganate with polycyclic-aromatic hydrocarbons (PAH), and then the acquired MnO_2_ were modified with a pH-responsive CaCO_3_ cover layer while ICG was simultaneously entrapped. PD-L1-targeting siRNA was loaded via electrostatic contact onto the positively charged Mn@CaCO_3_/ICG to generate the nanoplatform (Mn@CaCO_3_/ICG). In vivo studies have proven that the nanoplatform is capable of delivering the medicine to tumor tissues and improving tumor hypoxia, hence enhancing the therapeutic efficiency of photodynamic therapy. Additionally, the combinatorial effects of silencing the checkpoint gene PD-L1, which mediates immune evasion, lead to a startling therapeutic impact on rousing the immune system [146]. In summary, recent studies support the hypothesis that PDT is a highly effective method for producing ICD in cancer immunotherapy. Nevertheless, while the majority of research has used mouse models, clinical validation of this method is important. PDT and ICD constitute an exciting field of research with numerous potential applications in cancer treatment.

## 6. Conclusions and Future Perspectives

In recent years, immunotherapy has made great advancements in the treatment of various solid tumor types. However, for the majority of patients, a positive initial reaction to treatment diminishes over time, resulting in relapse and recurrence of cancer, limiting its clinical use. A critical component that contributes to the limited response to immunotherapies is the occurrence of numerous pathways regulating tumor immune suppression. Understanding the molecular mechanisms underpinning immune evasion could identify novel therapeutic targets for improving immunotherapy efficacy. Therefore, we reviewed the immune evasion mechanisms that contribute to the failure of cancer immunotherapy in this review.

The rapid growth of nanomedicine in recent years has provided fresh insights into cancer immunotherapy. Due to the development of effective delivery mechanism, the engineering of siRNA carriers has generated considerable interest, being capable of delivering siRNA into tumor tissues and tumor cell cytoplasm. siRNA has shown the ability to target any cancer-related genes, therefore establishing a new class of cancer therapies. The ideal nanocarrier device would shield the RNAi therapeutic drug from the vascular environment and deliver it to tumor cells efficiently. We believe that nano-siRNA medications have significant therapeutic potential in cancer treatment, and the significant progress in siRNA-based formulation will continue to expand our understanding of their therapeutic potential. However, various obstacles must be overcome before they can become trustworthy delivery systems. In part, the shortcomings of nano-siRNA in cancer immunotherapy are partially due to our limited insufficient understanding of the immune network during carcinogenesis. In light of the fact that innate and adaptive immunity comprise a complex network, the effect of depleting or suppressing a particular component on the network as a whole is still unknown. In the context of targeted drugs, suppression of one or more components may be compensated by the overexpression of other processes. Second, it is important to enhance modification of siRNA by nanocarriers to protect it from nuclease-based degradation. Additionally, nanoparticle toxicity assays are underdeveloped at the moment. Given the potential of nanoparticles’ physicochemical qualities to alter when they combine with various biological molecules in the body, their final forms should be carefully assessed. As a result, significant work is still required to improve the size, shape, ligands, and other features of nanoparticles, as well as to identify possible dangers, before they can be transferred into clinical practice. Endosomal escape strategies, cell and tissue targeting, and the creation of novel biomaterials are critical for the translation of siRNA from laboratory to clinical.

## Figures and Tables

**Figure 1 pharmaceutics-14-01344-f001:**
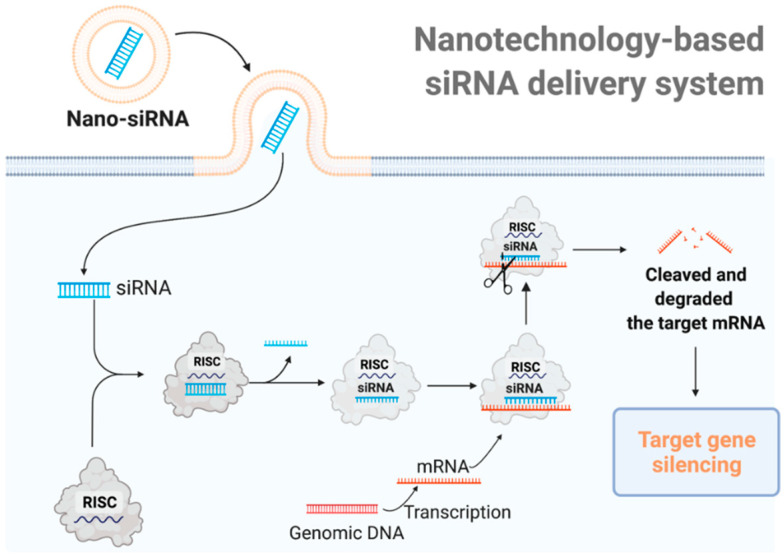
The mechanism of nanotechnology-based siRNA delivery systems. Utilizing delivery materials, siRNA can be delivered directly into the cell. The siRNA is integrated into the RNA-induced silencing complex (RISC) and the sense (passenger) strand is degraded by the RISC protein Argo-2. The remaining antisense strand acts as a guide for recognizing the complementary messenger RNA. The activated RISC–siRNA complex binds to and degrades the target mRNA, leading to the silence of the target gene.

**Figure 2 pharmaceutics-14-01344-f002:**
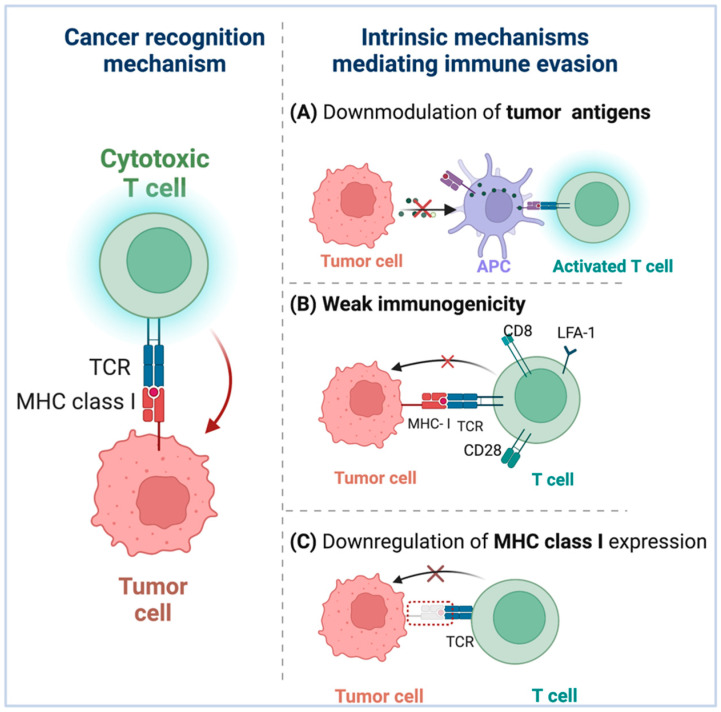
Tumor cells have evolved various strategies for evading immune responses. (**A**) Downregulation of tumor antigens: Certain malignancies lack pre-existing tumor T cell infiltration, allowing them to escape immunosurveillance due to low tumor antigen expression levels, resulting in inadequate APC recruitment and activation. (**B**) Weak immunogenicity: Immune selection permits tumors with weak immunogenicity to avoid immune surveillance and grow preferentially, and the weak immunogenicity of tumor antigens may be owing to incorrect or non-expression of costimulatory molecules on tumor cells. (**C**) Downregulation of MHC-I expression: By evading immune identification by tumor cells, the ability of tumor-associated antigen (TAA)-specific CTLs to kill cancer cells is compromised.

**Figure 3 pharmaceutics-14-01344-f003:**
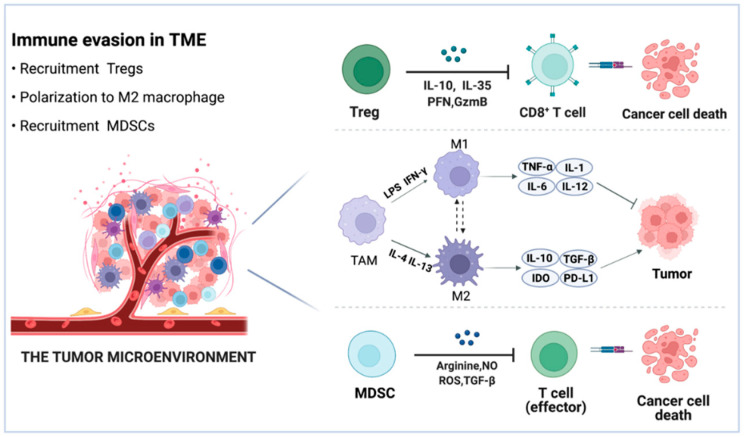
Schematic view of the TME. Tregs exert immunosuppressive effects via the release of IL10, IL-35, PFN, and GzmB. The preponderance of M2-like TAMs in the TME promotes tumor immune evasion. TAMs suppress the immune system in a variety of ways, including the release of IL-10 and TGF-β, activation of the IDO, and overexpression of the PD-L1 checkpoints. MDSCs limit CD8+ T cell and natural killer cell responses via arginine, NO, ROS, and TGF-β.

**Figure 4 pharmaceutics-14-01344-f004:**
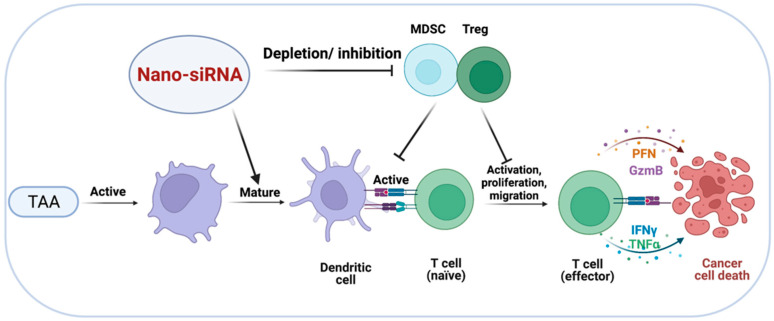
Targeted delivery of siRNA to MDSCs and Tregs. Depletion of Tregs and MDSCs or suppression of their immunosuppressive effects may restore the antitumor activity of effector T cells, resulting in cancer cells’ death.

**Figure 5 pharmaceutics-14-01344-f005:**
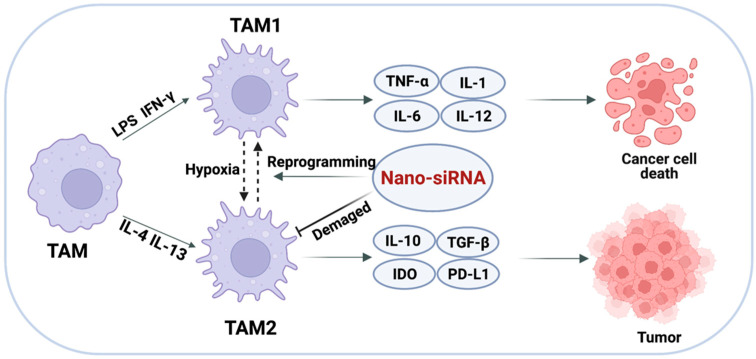
Targeted delivery of siRNA to TAMs. Altering the polarization of TAMs or impairing the survival and function of TAMs to achieve the elimination of immune evasion.

**Figure 6 pharmaceutics-14-01344-f006:**
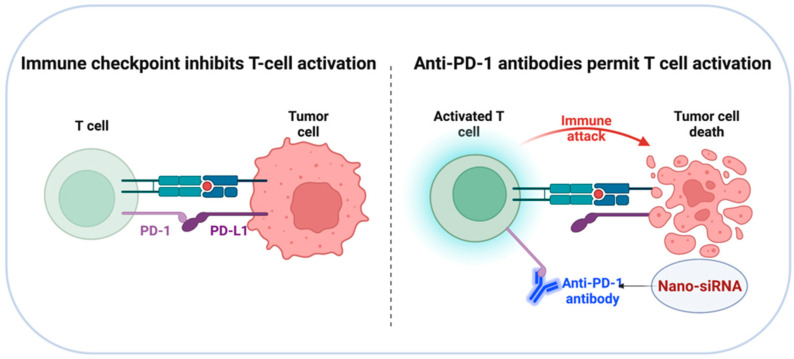
Diagrammatic illustration of immune checkpoint blockade therapies. Inhibiting the ligand–receptor interaction of PD-1/PD-L1 to prevent the activation of PD-1/PD-L1 signaling pathway, leading to T cell activation and reduction of tumor cell proliferation.

**Figure 7 pharmaceutics-14-01344-f007:**
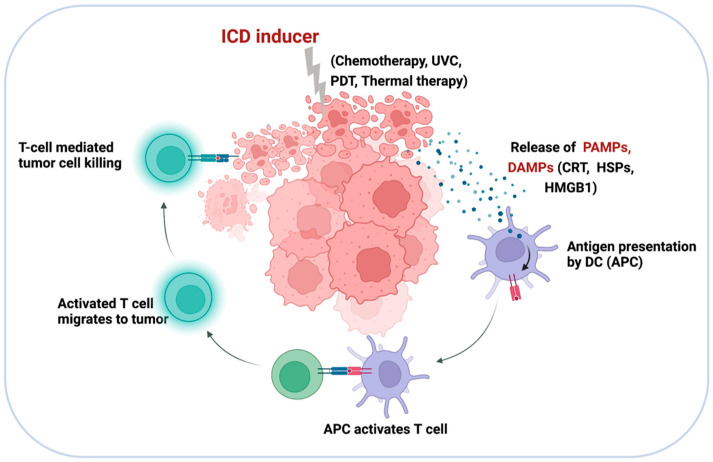
Diagrammatic illustration of the ICD mechanism. ICD-inducing strategies promote APC activation and priming of tumor-specific CD8+ T cells through the release of tumor antigens, PAMPs, and DAMPs, such as CRT, HSP70, HSP90, and HMG1.

## Data Availability

Not applicable.

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
