# Peer review of "Nanotechnology-Based siRNA Delivery Systems to Overcome Tumor Immune Evasion in Cancer Immunotherapy"

_pharmaceutics, 2022, doi:10.3390/pharmaceutics14071344_

Round 1

Reviewer 1 Report

Dear Professor Jamila Wang, 

The review manuscript "Nanotechnology-based siRNA delivery systems to overcome tumor immune evasion in cancer immunotherapy" by Dr. KaiLi Deng and co-workers reviewed and described about current advances in nanotechnology-based siRNA delivery strategies for overcoming immune evasion. In my opinion, this article is well organized, however, the details are not satisfactory. There are several comments should be responded before publication.

Major comments:

1.           The review is entitled "Nanotechnology-based siRNA delivery systems to overcome tumor immune evasion in cancer immunotherapy. While there is a detailed description of the mechanism after siRNA knockdown of the target gene, there is a lack of description of the crucial nanotechnology. The authors should explain more about the molecular design of siRNA carriers in the each section.

2.          Page 5, line 174     

What is “MDSC”? Note that often abbreviations are missing explanations.

3.          Page 9, line 268     

What is “CTLA4 ”?

4.          Page 8, line 298     

What is “CpG ODNs ”?

5.          Page 9, line 419     

What is “Mn@CaCO3/ICG ”?

Minor comment: 

Figure 1: “RISC (RNA-induced silencing complex)” is a complex of siRNA guide strand and protein. In this figure, the protein alone is referred to as “RISC”. This should be corrected.

Reviewer 2 Report

The proposed review "Nanotechnology-based siRNA delivery systems to overcome tumor immune evasion in cancer immunotherapy" is devoted to the problems of cancer therapy with application of siRNA base gene silencing strategy. Authors well described  the mechanims of tumor immune evasion and how they can be treated with application of siRNAs. The references are actual. The paper describes actual problems and could be interesting to very broad number of specialist in the area. In my opinion it could be published after minor revision.

Major comment: The type and composition of nanocarrier is imortant for intracellular penetration and escape from early endosomes. The role of nanocarrier chemistry on the success of strategy application in immunotherapy should be discussed throughout the text or in separate paragraph.

Minor comment: Page 1, line 28. Cancer seems to be on the second place in such a rank after heart diseases. 

Reviewer 3 Report

Dear Authors,

Re: [Nanotechnology-based siRNA delivery systems to overcome tumor immune evasion in cancer immunotherapy.  pharmaceutics-1742733

Title of the manuscript (which is in the form of a Review Article) brings the expectation to see an up-to-date literature review focused mainly on "nanocarriers" used for the delivery and targeting of siRNA.

The manuscript, however, lacks this main task and generally gives readers some Molecular Biology texts. There is absolutely no explanation of any of the nanocarriers employed for the encapsulation and delivery of siRNA. Only "exosomes" were mentioned though, very briefly. Perhaps you need to change the title of the article.

There are several punctuation errors (e.g. last words prior to References number: [38], [40] and [41] need to be separated from the brackets).

The sentence that includes the expression: "to escape evade killing by immune cells" needs to be re-written. 

Figures are excellent, however, they are too small and writings cannot be seen easily. 

Round 2

Reviewer 3 Report

Dear Authors,

Re: [Nanotechnology-based siRNA delivery systems to overcome tumor immune evasion in cancer immunotherapy.  pharmaceutics-1742733

Thanks for the revised version of your manuscript.